# Cancer Symptom Clusters in Children and Adolescents with Cancer Undergoing Chemotherapy: A Systematic Review

**DOI:** 10.3390/nursrep15050163

**Published:** 2025-05-07

**Authors:** Luís Carlos Lopes-Júnior, Jonathan Grassi, Marcela Bortoleto Freitas, Fernanda Ercília Souza Trigo, Fabrine Aguilar Jardim, Karolini Zuqui Nunes, Karla Anacleto de Vasconcelos, Regina Aparecida Garcia de Lima

**Affiliations:** 1Graduate Program in Nutrition and Health, Health Sciences Center at the Federal University of Espírito Santo (UFES), Vitória 29040-091, ES, Brazil; 2Graduate Program in Public Health, Health Sciences Center at the Federal University of Espírito Santo (UFES), Vitória 29040-091, ES, Brazil; 3Ribeirão Preto College of Nursing, University of São Paulo (USP), Ribeirão Preto 14040-902, SP, Brazil

**Keywords:** child, adolescent, neoplasms, antineoplastic agents, concurrent symptoms, oncology nursing

## Abstract

**Objective:** To synthesize and analyze the prevalence, composition, longitudinal stability, and predictors of cancer symptom clusters in children and adolescents undergoing chemotherapy. **Method:** A systematic literature review was conducted in accordance with the PRISMA 2020 guidelines. Evidence was sourced from MEDLINE/PubMed, Cochrane Library, Embase, PsycINFO, and Web of Science, as well as clinical trial registries (Clinical Trials WHO-ICTRP) and gray literature. The search was performed in February 2025, with no restrictions on publication date or language. Two independent reviewers screened and selected the studies. The methodological quality of the included studies was assessed using design-specific tools, and the findings were synthesized narratively. **Results:** A total of 6221 records were identified, with 12 studies meeting the inclusion criteria. These studies were published between 2010 and 2024 in the United States, Brazil, China, and Turkey. Cancer symptom clusters in children and adolescents followed well-defined patterns, with the gastrointestinal, emotional, fatigue-related, somatic, and self-image clusters being the most prevalent. **Conclusions:** Early identification of these cancer symptom clusters is essential for guiding interprofessional teams in delivering personalized, evidence-based care to children and adolescents with cancer and their families.

## 1. Introduction

The global age-standardized incidence, prevalence, and mortality rates of cancer in children and adolescents have declined significantly. Worldwide, 291,319 new cases and 98,834 deaths from childhood cancer have been reported, with a total of 1,806,630 prevalent cases [1]. Accurate data on the global burden of childhood and adolescent cancer are essential for informing international health policies and guiding resource allocation [2,3,4,5,6].

Pediatric cancer is often associated with symptoms that rarely occur in isolation but frequently manifest as cancer symptom clusters that are challenging to manage, including pain, nausea, vomiting, anorexia, cancer-related fatigue, depression, and anxiety, among others [7,8]. Both antineoplastic treatments and tumor progression contribute to the emergence of various symptom clusters during—and even after—treatment, adversely affecting functional status and significantly reducing quality of life [8,9,10]. The term “symptom cluster” refers to a group of interrelated symptoms that tend to occur together in a predictable pattern [11,12].

One of the key challenges in pediatric oncology nursing is the comprehensive assessment of oncological symptoms to demonstrate the significance of symptom clusters in terms of levels of interaction, association patterns, and synergistic effects [8]. The concept of a symptom cluster was first identified in murine models exposed to infectious processes and pro-inflammatory cytokines. These infectious and inflammatory responses led to the development of a phenomenon known as “sickness behavior”, a term introduced in 1992 [13,14] to describe a set of behavioral changes accompanying various pathological processes that appear to lack a direct pathophysiological link [14].

Similarly, symptoms such as fatigue, pain, sleep disturbances, depression, and cognitive impairment have been observed in children and adolescents with cancer, who exhibit elevated levels of pro-inflammatory cytokines [8,10,15].

Despite growing interest in understanding symptom clusters in oncology, existing systematic reviews have primarily focused on the adult population [16,17], or have provided only partial perspectives on pediatric oncology. While previous systematic reviews have explored aspects of symptom clusters in specific pediatric cancer contexts, they have not comprehensively synthesized evidence on the prevalence, composition, longitudinal stability, and predictors of symptom clusters in children and adolescents undergoing chemotherapy. For instance, a recent scoping review focusing on gastrointestinal symptom clusters in pediatric oncology, providing valuable insights but lacking a systematic synthesis of broader symptom clusters and their interrelationships [18]. Similarly, other researchers reviewed self-reported symptoms in children under 8 years of age, but their scope was limited to younger pediatric patients and did not analyze symptom clustering patterns in a broader age range [19]. Additionally, symptom clusters were reviewed in adolescents, but it was not a systematic review and did not examine longitudinal stability or predictive factors that influence symptom cluster trajectories [20].

Given these limitations, the present systematic review fills a critical gap by synthesizing robust evidence on cancer symptom clusters in pediatric and adolescent populations. Understanding how these clusters evolve over time and identifying predictive factors for co-occurring symptoms could provide a scientific foundation for developing personalized therapeutic interventions. Furthermore, this approach may enable the early identification of symptom patterns predictive of significant symptom clusters, ultimately contributing to improved symptom management and better quality of life during and after cancer treatment.

The aim of this study was to synthesize and analyze the prevalence, composition, longitudinal stability, and predictors of cancer symptom clusters in children and adolescents undergoing chemotherapy. By addressing these knowledge gaps, our findings have the potential to inform targeted clinical interventions and enhance supportive care strategies in pediatric oncology.

## 2. Method

### 2.1. Study Design

This study is a systematic literature review, conducted in compliance with the Preferred Reporting Items for Systematic Reviews and Meta-Analyses (PRISMA) 2020 guidelines [21]. Additionally, the review protocol was registered with the International Prospective Register of Systematic Reviews (PROSPERO), under registration number CRD4202283141, and the protocol article was subsequently published in another journal [22].

### 2.2. Research Question

The research question was formulated using the PECO framework (Patient/Population, Exposure, Control or Comparison, Outcomes) [23], where P (Population) = children and adolescents (0 to 19 years) diagnosed with malignant neoplasms, E (Exposure) = chemotherapy treatment, C (Comparison) = not applicable, and O (outcomes) = prevalence, composition, longitudinal stability, and predictors of cancer symptom clusters. Based on the PECO acronym, the following guiding research question was established: “What scientific evidence is available regarding the prevalence, composition, longitudinal stability, and predictors of cancer symptom clusters in children and adolescents with cancer undergoing chemotherapy”?

### 2.3. Eligibility Criteria

All quantitative studies (observational and experimental studies) involving pediatric and adolescent cancer patients (0 to 19 years old) of both sexes, undergoing chemotherapy and diagnosed with any type of malignant neoplasm, were included. No restrictions on publication date or language were applied in the database search strategy. Qualitative studies and experimental studies conducted in animal models, in vivo or ex vivo, were excluded.

### 2.4. Search Strategy

The evidence search was conducted across the following databases: Medical Literature Analysis and Retrieval System Online (MEDLINE) via PubMed, Cochrane Library, Excerpta Medica Database (Embase), Psychology Information Database (PsycINFO), and Web of Science. Additionally, trial registry platforms such as ClinicalTrials.gov and the WHO International Clinical Trials Registry Platform were included. To enhance comprehensiveness, gray literature sources were also searched, including The British Library (UK), Google Scholar, and Preprints for Health Sciences (medRxiv). The search strategy combined controlled descriptors (database-specific indexing terms) and keywords. For MEDLINE, Medical Subject Headings (MeSH) were used; for Embase, Emtree terms were applied; and for PsycINFO, PsycINFO Thesaurus was consulted, with Boolean operators (AND/OR) used to refine the search. Table 1 provides a detailed and transparent overview of the search strategy employed in this systematic review.

The Boolean operators “AND” and “OR” were used to achieve restrictive and additive combinations, respectively. Furthermore, the search was conducted using broadly expanded descriptors, without applying database filters, to ensure a comprehensive sample and minimize the risk of data loss.

### 2.5. Screening of Articles

Following the search, articles were exported to EndNote Web™ for storage, organization, and reference management, including duplicate identification. The retrieved studies were then exported to the Rayyan QCRI app, a screening and selection platform developed by the Qatar Computing Research Institute [24]. During the initial screening, titles and abstracts were independently reviewed by two researchers (JG and MBF) using Rayyan™. After this preliminary selection, the same two researchers independently assessed the full texts of the eligible studies to determine inclusion or exclusion. To resolve disagreements, a third reviewer, an expert in systematic review methodology (LCLJ), was consulted for arbitration in cases where the initial reviewers disagreed on study inclusion.

### 2.6. Data Extraction

Two reviewers, as previously mentioned, independently extracted data from the included studies using predefined data extraction forms [25,26,27]. The extracted information included the following: (a) study identification, including article title, journal impact factor, authors’ countries, year of publication, host institution (hospital, university, research center, multicenter, or single-institution study), and conflicts of interest and funding sources; (b) methodological characteristics (study objective, research question or hypotheses, study design, sample size, age, baseline characteristics of patients, recruitment method, attrition, outcome measures, follow-up duration, and statistical analyses); (c) main results and implications for clinical practice; and (d) study conclusions.

### 2.7. Methodological Quality Assessment

To classify the selected studies, we applied the hierarchy of evidence from the Center for Evidence-Based Medicine [28], which organizes evidence into five hierarchical levels based on study design (1A, 1B, 1C, 2A, 2B, 2C, 3A, 3B, 4, and 5). The methodological quality of studies was independently assessed by two independent reviewers, with a third reviewer resolving any disagreements. For quasi-experimental studies, the Risk of Bias in Non-randomized Studies of Interventions (ROBINS-I) tool was used [29], which evaluates bias risk across seven domains, grouped into three dimensions: pre-intervention, intervention, and post-intervention. The risk of bias is categorized as low, moderate, serious, or critical [29]. The overall ROBINS-I assessment is assigned as follows: (a) low risk of bias: comparable to a well-designed randomized trial (low bias risk across all domains); (b) moderate risk of bias: consistent with a non-randomized trial design, though not comparable to a well-designed randomized trial (low or moderate risk across all domains); (c) serious risk of bias: significant concerns in at least one domain (low or moderate risk across most domains, with at least one domain showing serious bias); (d) critical risk of bias: too problematic to provide evidence (critical risk in at least one domain); and (e) no information: insufficient data for any bias risk judgment in one or more domains [29]. For cross-sectional and longitudinal studies, we applied the JBI Critical Appraisal Checklist, following JBI recommendations [30]. The results of the included studies are presented narratively.

## 3. Results

### 3.1. Study Selection

A total of 6221 records were retrieved from electronic databases and registries. After removing 420 duplicates using EndNote™, 5802 studies proceeded to the title and abstract screening phase. During this stage, conducted via Rayyan™, 5570 studies were excluded for not meeting the predefined inclusion criteria. This screening process resulted in 32 studies being selected for full-text review. Following the full-text assessment, an additional 20 studies were excluded (12 due to different outcomes and 8 based on study design), leading to the inclusion of 12 articles for qualitative synthesis and analysis, as illustrated in Figure 1.

### 3.2. Characterization of Included Studies

Among the twelve included studies [31,32,33,34,35,36,37,38,39,40,41,42], publication dates ranged from 2010 to 2024, with all studies published in English in four different countries (China, United States, Turkey, and Brazil) [31,32,33,34,35,36,37,38,39,40,41,42], six studies were cross-sectional observational designs [32,33,34,37,40,41], while four were quasi-experimental studies [31,35,36,38], and two were longitudinal studies [39,42]. Table 2 chronologically summarizes the main characteristics of the studies included in the qualitative synthesis.

This systematic review identified several symptom clusters commonly observed among children and adolescents undergoing cancer treatment. The analyzed studies indicate that these clusters vary based on cancer type, treatment phase, and individual factors, yet they exhibit consistent patterns.

1.Gastrointestinal Cluster

The gastrointestinal cluster was one of the most frequently reported across multiple studies, encompassing symptoms such as nausea and vomiting (commonly associated with chemotherapy); diarrhea and constipation; taste alterations and dry mouth; and loss of appetite and weight loss. Studies by Li et al. [37,40,42], Wang et al. [41], and Atay et al. [33] emphasized the significance of this cluster, linking it to specific phases of chemotherapy and its adverse impact on quality of life.

2.Emotional and Psychosocial Cluster

Another widely reported cluster involved emotional and psychological symptoms, including anxiety, worry, and nervousness; irritability and sadness; depression; and altered self-perception. This cluster was identified in several studies [31,33,35,37,39], highlighting the importance of psychological support throughout cancer treatment.

3.Fatigue and Sleep Disturbance Cluster

Fatigue-related symptoms were highly prevalent, significantly affecting patients’ physical and emotional well-being. Key symptoms included severe and persistent fatigue; sleep disturbances and insomnia (or psychological stress); and reduced physical activity levels. This cluster was highlighted by Hockenberry et al. [31,35], Lopes-Júnior et al. [36,38], and Hooke et al. [39], with greater severity observed in adolescents diagnosed with solid tumors and those undergoing chemotherapy.

4.Somatic and Neurological Cluster

This cluster comprised physical symptoms associated with cancer treatment, such as muscle pain and tingling sensations; headache and dizziness; and fever and excessive sweating. Studies by Li et al. [37,40,42], and Wang et al. [41] identified this cluster primarily in patients diagnosed with acute lymphoblastic leukemia (ALL).

5.Self-Image and Dermatological Cluster

Several studies reported symptoms associated with treatment-related changes in physical appearance, including hair loss (alopecia); skin changes (dryness, itching); and swelling in the extremities. This cluster was reported by Li et al. [37,40,42], and Atay et al. [33], emphasizing the psychosocial impact of self-image alterations on pediatric oncology patients.

### 3.3. Internal Validity of Included Studies

Most of the quasi-experimental studies (n = 3; 75%) [33] were classified as having a low overall risk of bias, according to the ROBINS-I tool criteria. Regarding the risk of bias in the selected cross-sectional studies, which were evaluated using the JBI Critical Appraisal Checklist, all (n = 7; 87.5%) were classified as having good methodological quality. Table 3 and Table 4 provide a detailed assessment of the methodological quality of the quasi-experimental and cross-sectional studies, respectively.

## 4. Discussion

The studies included in this review consolidate evidence on the prevalence, composition, longitudinal stability, and predictors of oncology symptom clusters in children and adolescents undergoing chemotherapy treatment. Across the studies, a total of 1099 participants aged 0 to 19 years were assessed. Among the symptom clusters evaluated, fatigue was the most frequently reported in all studies, followed by sleep disturbances, loss of appetite, nausea, and vomiting, respectively. In addition, five clusters were found to be more prevalent: gastrointestinal, emotional, fatigue-related, somatic, and self-image.

The instruments most commonly used for assessing oncology symptom clusters in the pediatric population were the Memorial Symptom Assessment Scale (MSAS), the Lansky Play-Performance Scale (LPPS), the Karnofsky Performance Status Scale (KPS), the Childhood Fatigue Scale (CFS), and the Pediatric Quality of Life Multidimensional Fatigue Scale (PedsQL MFS). In the methodological quality assessment, 10 studies (83.3%) demonstrated a low risk of bias, reinforcing the strong methodological rigor of the studies included in this review.

Cancer-related fatigue (CRF) is defined as a distressing and pervasive symptom with physical, mental, and emotional components, characterized by a lack of energy. It is recognized as the most common symptom experienced by children and adolescents with cancer and differs significantly from the fatigue found in the general population [39,42,43,44,45].

Among children and adolescents undergoing chemotherapy, the prevalence of cancer-related fatigue ranges from 50% to 93% [41,46,47]. Notably, CRF is not easily alleviated by sleep or rest and may persist for months or even years following the completion of cancer treatment [41,48,49]. Previous reviews have highlighted fatigue as a highly prevalent and distressing symptom that significantly impairs quality of life, becoming a barrier to daily activities, which are an essential part of childhood and adolescence [49,50,51,52,53,54,55,56]. Moreover, the use of certain medications in treatment can lead to metabolic alterations, directly affecting daily functioning. This impact is particularly pronounced in the immune system, especially through the pro-inflammatory cytokine-mediated signaling pathways [14].

A study assessing symptoms experienced by children with cancer at home identified fatigue (52.1%), nausea (50.7%), loss of appetite (43.7%), and pain (42.3%) as the most frequent physical symptoms. The most common psychological symptoms included sleep difficulties (21.1%), worry (18.3%), sadness (18.3%), and nervousness (16.9%). Significant differences were found in overall physical and psychosocial symptoms, as well as in the Global Distress Index, between patients with and without pain, fatigue, and nausea [57].

Fatigue is often regarded as an underlying driver of other distressing symptoms, given its potential to be intense, disabling, distressing, and significantly depressive [58]. Fatigue, sleep disturbance, pain, and depression are interrelated and frequently co-occur, exacerbating one another and further compromising quality of life during chemotherapy [59]. The presence of multiple concurrent physical symptoms, combined with psychological distress, leads to substantial impairment, directly interfering with daily functioning and hindering the patient’s ability to complete the prescribed cancer treatment [60].

Depression is bidirectionally associated with greater physical multimorbidity [61]. The prevalence of depression in cancer patients is 3.4 times higher than in the general population [62]. Given the high rates of anxiety and depression among children with cancer [63], emotional distress should be recognized as a sixth vital sign and routinely assessed in all cancer patients [64]. Moreover, the most significant somatic associations with depression include comorbiditiesrelated to the disease, the symptom burden related to cancer, and pain [39,65].

Sleep disturbances are common in pediatric patients [66] and frequently co-occur within the same cluster as cancer-related fatigue [39,58,67]. Sleep requirements change with age, with adolescents typically needing fewer hours of sleep than younger children [68]. These disturbances are not limited to the treatment period but can persist for more than five years following therapy [69]. Chemotherapeutic agents, steroids, and radiation therapy, which are the cornerstones of pediatric cancer treatment, play a significant role in disrupting sleep patterns[70]. A reciprocal relationship exists between sleep and pain, where pain leads to sleep disturbances and poor sleep increases pain perception [71]. Additionally, the cancer diagnosis, treatment-related consequences, and other risk factors may trigger sleep disturbances in cancer patients [66]. Hospitalization is another key factor influencing sleep quality in pediatric cancer patients. While outpatients tend to experience better sleep quality than hospitalized patients, they still face an increased risk of sleep disturbances during active treatment [71].

The quality of life of pediatric patients can be significantly affected by psychological stress [39,40,43,45,58]. As treatment progresses, critical aspects of the immune response may be suppressed, including natural killer (NK) cell activity and T cell proliferation [72]. However, psychological stress related to cancer may be alleviated through targeted interventions, such as therapeutic clowning [37].

Chemotherapy primarily targets DNA and protein expression in both cancer cells and normal host cells, resulting in a narrow therapeutic index and high risk of toxicity. Most chemotherapeutic agents affect rapidly dividing cells, particularly those in tissues with high cellular turnover, including the bone marrow, gastrointestinal tract, and hair follicles. Common toxicities associated with chemotherapy include myelosuppression, mucositis, nausea, vomiting, weight loss, diarrhea, alopecia, fatigue, among others. As a result, immunosuppression significantly increases the risk of infections [73].

The number and composition of symptom clusters in the studies included in this review varied during the first three months of treatment [34]. The most prevalent symptoms identified were dizziness, irritability, fatigue [32,34,36,37,43], nausea [32,34,36], depression [34,36], vomiting [32,34], weight loss [32,34], sleep disturbances [36], pain [36], sadness [36], anxiety-related worry [34], and psychological stress [37,43].

Although chemotherapy has led to continuous advancements in treatment, significantly improving overall survival, patients experience a broad range of physical and psychological symptoms that impact quality of life [56].

Gastrointestinal symptoms are highly prevalent among children and adolescents with cancer as well as their parents [41,74,75]. This finding is consistent with the results of this systematic review, which also identified gastrointestinal symptom cluster as one of the most prevalent in this population.

The term “gastrointestinal symptoms” encompasses symptoms directly associated with the structures and functions of the gastrointestinal tract, including dry mouth, mouth sores, difficulty swallowing, sore throat, jaw pain, nausea, vomiting, loss of appetite, changes in taste and food sensations, weight loss, diarrhea, and constipation [40,41,76,77]. Together, these symptoms contribute to feeding difficulties, which can lead to malnutrition and subsequently worsen clinical outcomes [78]. Children diagnosed with malnutrition three months after cancer diagnosis have significantly higher rates of febrile neutropenia during the first year following diagnosis and lower survival rates [79].

Loss of appetite, often induced by cancer and chemotherapy, is commonly attributed to changes in taste, mouth sores, nausea, and vomiting. These factors exacerbate other symptoms, including pain, fatigue, depression, and anxiety [80]. Functional decline associated with nutritional intake is often linked to progressive weight loss. In addition, mouth sores, pain, and poor nutrition can increase the risk of infection in immunocompromised patients [80].

A recent scoping review synthesizing findings from studies on the gastrointestinal symptom cluster and associated non-gastrointestinal symptoms in children undergoing cancer treatment revealed 12 of the most frequently reported symptoms across all clusters: nausea, changes in taste, vomiting, loss of appetite, weight loss, constipation, dry mouth, diarrhea, mouth sores, fatigue, changes in appearance, and hair loss [80].

Longitudinal stability refers to the long-term trajectory and progression of a symptom, allowing for the identification of patients’ primary needs and enabling better prevention and health promotion [81]. In children and adolescents with cancer undergoing chemotherapy with cisplatin, doxorubicin, or ifosfamide, symptoms such as cancer-related fatigue, depression, and compromised performance status can emerge as early as the first day of therapy [32]. One of the key factors influencing oncology symptom clusters during chemotherapy is the treatment stage at which patients are currently undergoing therapy [32].

The gastrointestinal and emotional symptom clusters are among the most prevalent in pediatric oncology and can persist beyond the chemotherapy phase [32,33]. Children and adolescents undergoing chemotherapy experience a wide range of persistent symptoms, with greater distress observed among those actively receiving treatment [33]. Notably, chemotherapy not only influences the formation of symptom clusters but also affects their intensity and severity. As chemotherapy cycles progress, symptom clusters in children and adolescents tend to become more intense and severe [36,38]. A comprehensive understanding of symptom clusters across different phases of cancer treatment is essential for identifying symptom stability across patient subgroups by age and over time. Such insights can inform timely clinical decisions, ultimately helping to alleviate symptom burden in children and adolescents with cancer.

It is noteworthy that several approaches have been used to identify clusters in children and adolescents with cancer, including a priori approaches (focused assessment of symptoms known to be prevalent in a particular clinical population) and de novo approaches (assessment of a wider range of symptoms to identify clusters through robust statistical analyses) [82,83]. Most studies utilizing the de novo approach applied advanced statistical techniques to identify new clusters [34,35,84]. The number of symptom clusters reported in previous studies has ranged from 1 to 10, with some inconsistencies in the composition of clusters containing similar symptoms [84,85]. Despite these inconsistencies, insights gained from symptom clusters enhance our understanding of underlying mechanisms, which could serve as potential targets for intervention [86].

The multidisciplinary team plays a crucial role in promoting the health of pediatric oncology patients [87]. Recognizing and implementing early interventions for symptom clusters is an essential component of comprehensive care.

Children and adolescents with cancer face significant challenges at individual, family, and social levels due to the disease and its treatment-related effects. This underscores the need for patient- and family-centered care [88], aimed at fostering hope and trust, providing education and support, and addressing the child’s complex needs throughout the therapeutic journey.

Parents play a crucial role in the well-being of pediatric oncology patients from the beginning of their therapeutic journey. Establishing a trusting partnership between parents and the healthcare team is essential to ensuring effective patient care. This collaboration helps parents recognize their fundamental role, encouraging their active participation in their child’s care and keeping them informed about disease progression and treatment plans [89]. Additionally, understanding the experiences of family caregivers can help healthcare professionals plan care strategies and develop interventions to mitigate symptom clusters [89,90].

A notable limitation of this review is the exclusion of certain databases, such as Scopus and CINAHL, which may have omitted relevant publications that could enhance the validity and comprehensiveness of the findings.

The experience of a cancer diagnosis is often traumatic and highly stressful for children and adolescents. The emotional, psychological, and social burdens associated with cancer coping mechanisms in this age group are significant and warrant greater attention, particularly in the context of cancer symptom clusters. Coping with cancer in childhood and adolescence presents a complex challenge that necessitates a holistic approach, integrating both clinical and psychosocial aspects of care. Providing adequate support is essential for minimizing psychosocial distress and enhancing overall well-being. Finally, the ability to predict symptom clusters in children and adolescents undergoing chemotherapy can aid in treatment decision-making, helping to alleviate symptom burden and improve quality of life in this population [38].

## 5. Conclusions

Findings from this systematic review indicate that cancer symptom clusters in children and adolescents follow distinct, well-defined patterns, with the gastrointestinal, emotional, fatigue, somatic, and self-image clusters being the most prevalent. Identifying these clusters is essential for developing targeted interventions to alleviate symptom burden and improve quality of life in pediatric oncology patients. The longitudinal stability of cancer symptom clusters varies depending on the disease stage and treatment phase, with inflammatory mediators, particularly pro-inflammatory and anti-inflammatory cytokines, serving as key predictors of symptom cluster configurations. Early recognition of these cancer symptom clusters is critical for guiding interdisciplinary teams in delivering personalized, patient-centered care for children and adolescents with cancer and their families.

## Figures and Tables

**Figure 1 nursrep-15-00163-f001:**
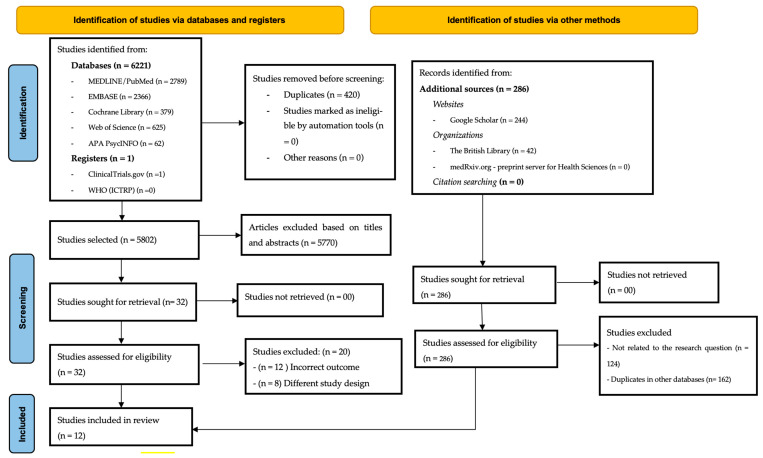
PRISMA flowchart for study selection.

**Table 1 nursrep-15-00163-t001:** Search strategy used in electronic databases and additional sources, on 10 February 2025. Vitória, ES, Brazil.

Database	Search Strategy
MEDLINE/PubMed	#1 (“Infant, Newborn” [MeSH Terms] OR “Infants, Newborn” [Title/Abstract] OR “Newborn Infant” [Title/Abstract] OR “Newborn Infants” [Title/Abstract] OR “Newborns” [Title/Abstract] OR “Newborn” [Title/Abstract] OR “Neonate” [Title/Abstract] OR “Neonates” [Title/Abstract] OR “Infant” [MeSH Terms] OR “Infants” [Title/Abstract] OR “Child” [MeSH Terms] OR “Children” [Title/Abstract] OR “Child, Preschool” [MeSH Terms] OR “Preschool Child” [Title/Abstract] OR “Children, Preschool” [Title/Abstract] OR “Preschool Children” [Title/Abstract] OR “Adolescent” [MeSH Terms] OR “Adolescents” [Title/Abstract] OR “Teens” [Title/Abstract] OR “Teenagers” [Title/Abstract] OR “Teenager” [Title/Abstract] OR “Youth” [Title/Abstract] OR “Youths” [Title/Abstract]) #2 (“Neoplasms” [MeSH Terms] OR “Neoplasia” [Title/Abstract] OR “Neoplasias” [Title/Abstract] OR “Neoplasm” [Title/Abstract] OR “Tumors” [Title/Abstract] OR “Tumor” [Title/Abstract] OR “Cancer” [Title/Abstract] OR “Cancers” [Title/Abstract] OR “Malignancy” [Title/Abstract] OR “Malignancies” [Title/Abstract] OR “Malignant Neoplasms” [Title/Abstract] OR “Malignant Neoplasm” [Title/Abstract] OR “Neoplasm, Malignant” [Title/Abstract] OR “Neoplasms, Malignant” [Title/Abstract]) #3 #1 AND #2 #4 (“Chemotherapy” [Title/Abstract] OR “Induction Chemotherapy” [MeSH Terms] OR “Chemotherapy, Induction” OR “Induction Chemotherapies” [Title/Abstract] OR “Consolidation Chemotherapy” [MeSH Terms] OR “Chemotherapy, Consolidation” [Title/Abstract] OR “Consolidation Chemotherapies” [Title/Abstract] OR “Maintenance Chemotherapy” [MeSH Terms] OR “Chemotherapy, Maintenance” [Title/Abstract] OR “Maintenance Chemotherapies” [Title/Abstract] NOT “Radiotherapy” [MeSH Terms] NOT “Surgery” [Subheading]) #5 (“Signs and Symptoms” [MeSH Terms] OR “Symptoms and Signs” [Title/Abstract] OR “Symptom Cluster” [Title/Abstract] OR “Cluster, Symptom” [Title/Abstract] OR “Clusters, Symptom” [Title/Abstract] OR “Symptom Clusters” [Title/Abstract] OR “Cancer Symptom Clusters” [Title/Abstract] OR “Symptom Constellation” [Title/Abstract] OR “Symptom Management” [Title/Abstract]) #6 #3 AND #4 AND #5
Cochrane Library	#1 (Infant, Newborn) OR (Infants, Newborn) OR (Newborn Infant) OR (Newborn Infants) OR (Newborns) OR (Newborn) OR (Neonate) OR (Neonates) OR (Infant) OR (Infants) OR (Child) OR (Children) OR (Child, Preschool) OR (Preschool Child) OR (Children, Preschool) OR (Preschool Children) OR (Adolescent) OR (Adolescent) OR (Adolescents) OR (Teens) OR (Teenagers) OR (Teenager) OR (Youth) OR (Youths) #2 (Neoplasms) OR (Neoplasia) OR (Neoplasias) OR (Neoplasm) OR (Tumors) OR (Tumor) OR (Cancer) OR (Cancers) OR (Malignancy) OR (Malignancies) OR (Malignant Neoplasms) OR (Malignant Neoplasm) OR (Neoplasm, Malignant) OR (Neoplasms, Malignant) #3 #1 AND #2 #4 (Chemotherapy) OR (Induction Chemotherapy) OR (Chemotherapies, Induction) OR (Chemotherapy, Induction) OR (Induction Chemotherapies) OR (Consolidation Chemotherapy) OR (Chemotherapy, Consolidation) OR (Consolidation Chemotherapies) OR (Maintenance Chemotherapy) OR (Chemotherapy, Maintenance) OR (Maintenance Chemotherapies) NOT (Radiotherapy) NOT (Surgery) #5 (Signs and Symptoms) OR (Symptoms and Signs) OR (Symptom Cluster) OR (Cluster, Symptom) OR (Clusters, Symptom) OR (Symptom Clusters) OR (Cancer Symptom Clusters) OR (Symptom Constellation) OR (Symptom Management) #6 #3 AND #4 AND #5
EMBASE	#1 (‘child’/exp OR ‘preschool child’/exp OR ‘child, preschool’ OR ‘pre-school child’ OR ‘pre-school going children’ OR ‘pre-schooler’ OR ‘pre-schoolers’ OR ‘preschool child institution’ OR ‘preschooler’ OR ‘adolescent’/exp OR ‘teenager’ OR ‘juvenile’ OR ‘youth’) #2 (‘neoplasm’/exp OR ‘tumor’ OR ‘tumour’ OR ‘tumors’ OR ‘malignant neoplasm’ OR ‘cancer’ OR ‘cancers’ OR ‘malignant neoplasia’ OR ‘malignant neoplastic disease’ OR ‘malignant tumor’ OR ‘malignant tumour’ OR ‘neoplasia, malignant’ OR ‘tumor, malignant’ OR ‘tumour, malignant’) #3 (‘chemotherapy’/exp OR ‘induction chemotherapy’/exp OR ‘consolidation chemotherapy’/exp OR ‘maintenance chemotherapy’ OR ‘chemotherapeutics’ OR ‘chemotherapy, induction’ OR ‘chemotherapeutic consolidation’ OR ‘ chemotherapy consolidation’) #4 (‘physical disease by body function’/exp OR ‘symptom’/exp) #5 #1 AND #2 AND #3 AND #4
Web of Science	#1 ALL = ((Infant, Newborn) OR (Infants, Newborn) OR (Newborn Infant) OR (Newborn Infants) OR (Newborns) OR (Newborn) OR (Neonate) OR (Neonates) OR (Infant) OR (Infants) OR (Child) OR (Children) OR (Child, Preschool) OR (Preschool Child) OR (Children, Preschool) OR (Preschool Children) OR (Adolescent) OR (Adolescent) OR (Adolescents) OR (Teens) OR (Teenagers) OR (Teenager) OR (Youth) OR (Youths)) #2 ALL = ((Neoplasms) OR (Neoplasia) OR (Neoplasias) OR (Neoplasm) OR (Tumors) OR (Tumor) OR (Cancer) OR (Cancers) OR (Malignancy) OR (Malignancies) OR (Malignant Neoplasms) OR (Malignant Neoplasm) OR (Neoplasm, Malignant) OR (Neoplasms, Malignant)) #3 #1 AND #2 #4 ALL = ((Chemotherapy) OR (Chemotherapy, Adjuvant) OR (Induction Chemotherapy) OR (Consolidation Chemotherapy) OR (Maintenance Chemotherapy)) #5 ALL = ((Symptom Cluster) OR (Cluster, Symptom) OR (Clusters, Symptom) OR (Symptom Clusters) OR (Cancer Symptom Clusters) OR (Symptom Constellation) OR (Symptom Management)) #6 #3 AND #4 AND #5
PsycINFO	#1 (Adolescent Health OR Preschool Students OR Child Health) #2 (Neoplasms OR Malignant Neoplasms OR Cancers OR Carcinomas OR Tumors) #3 (Chemotherapy OR Chemotherapeutics) #4 (Symptoms OR Symptoms Based Treatment OR Symptom Cluster OR Cancer Symptom Cluster) #5 #1 AND #2 AND #3 AND #4
Clinicaltrial.gov	#1 (Child) OR (Children) OR (Adolescent) OR (Adolescents) #2 (Neoplasms) OR (Neoplasia) #3 (Chemotherapy) #4 (Symptom Cluster) OR (Symptom Management) #5 #1 AND #2 AND #3 AND #4
WHO International Clinical Trials Registry Platform	(Child OR Children OR Adolescent OR Adolescents) AND (Neoplasms OR Neoplasia) AND (Chemotherapy) AND (Symptom Cluster OR Symptom Management)
The British Library (UK)	(Child OR Children OR Adolescent OR Adolescents) AND (Neoplasms OR Neoplasia) AND (Chemotherapy) AND (Symptom Cluster OR Symptom Management)
Google Scholar	(Children OR Adolescents) AND (Neoplasms OR Neoplasia) AND (Chemotherapy) AND (Symptom Cluster)
Preprints in Health Sciences (medRxiv)	(Children OR Adolescents) AND (Neoplasms OR Neoplasia) AND (Chemotherapy) AND (Symptom Cluster)

**Table 2 nursrep-15-00163-t002:** Characteristics of the studies included in the systematic review.

Reference/ Country	Design/ Level of Evidence	Objective	Sample	Symptom Assessed/ Instruments	Main Results
Hockenberry et al., 2010 [31] United States	Quasi-experimental 2B	To examine the influence of CSC fatigue, nausea/vomiting, and sleep disturbances on clinical outcomes and behavior in children and adolescents, before and after receiving chemotherapy	67 children and teenagers Age: 7 to 18 years (average age: 12.3 years)	Symptoms: Fatigue, nausea/vomiting, performance status, depression and sleep disturbances Instruments: CFS/ AFS/PFS/wrist actigraph/CDI/BASC/LPPS/KPS	When analyzed as a symptom cluster, fatigue, sleep disturbances, nausea, and vomiting had a negative impact on depressive symptoms and behavioral changes in adolescents after chemotherapy. In children, however, fatigue alone influenced depressive symptoms and behavioral changes. Fatigue was a significant predictor of depressive symptoms in children (F[1,67] = 18.427, *p* < 0.001). Additionally, higher parental perception of fatigue in their children was associated with greater reported behavioral and emotional difficulties (F[1,89] = 12.535, *p* = 0.001).
Atay 2011 [32] Turkey	Cross-sectional 3B	To describe the prevalence of CSC symptoms that occur in children and adolescents receiving chemotherapy or who have completed treatment	164 children and adolescents (79 boys and 85 girls) Age: 10 to 18 years (mean age: 13.9 ± 2.1 years)	Symptoms: 30 cancer symptoms Instrument: MSAS	5 CSC were identified: Cluster 1: trouble urinating, diarrhea, difficulty swallowing, constipation. Cluster 2: dyspnea, dizziness, mouth sores, taste changes, weight loss, dry mouth. Cluster 3: lack of concentration, skin changes, numbness/tingling, swelling of arms/legs, itching, insomnia, sweating, coughing. Cluster 4: worried, irritated, “I don’t look like myself”, hair loss. Cluster 5: nausea, vomiting, fatigue. The most prevalent CSC included nervousness, sadness, and fatigue. The most distressing symptoms in patients aged 10 to 18 y.o. were nausea/vomiting and alopecia.
Atay et al. 2012 [33] Turkey	Cross-sectional 3B	To determine the prevalence of CSC in children at 3, 6, and 12 months after cancer diagnosis	44 adolescents and children (30 boys and 24 girls) Mean age: 14 years	Symptoms: 30 symptoms were assessed for severity and distress Instrument: MSAS	Four CSC were identified within the first month after diagnosis: Cluster 1: Dizziness, changes in food taste, and worry. Cluster 2: Feeling irritable, sad, and nervous. Cluster 3: Vomiting, nausea, and lack of energy. Cluster 4: Sweating, diarrhea, and insomnia.
Baggott et al., 2012 [34] United States	Cross-sectional 3B	To compare the number and types of CSC identified through patient ratings of symptoms	131 children and adolescents Age: 10 to 18 years	Symptoms: Chemotherapy sequelae, mood disorders and neuropsychological distress Instruments: MSAS/KPS	Most symptoms contributed similarly to the chemotherapy sequelae cluster. For mood disturbance cluster, feeling irritable made the weakest contribution. In the neuropsychological distress cluster, feeling irritable, altered self-perception, and skin changes were the weakest contributors (χ^2^ = 322.7, *p* < 0.01).
Hockenberry et al., 2011 [35] United States	Quasi-experimental 2B	To examine the effects of synergistic symptoms experienced by pediatric patients during cancer therapy	67 children and adolescents (38 boys and 29 girls) Age: 7 to 18 years (average age: 12.3 years)	Symptoms: Fatigue, nausea and vomiting, depression, and performance status symptoms Instruments: CFS/ AFS/PFS/wrist actigraph/CDI/BASC/LPPS/KPS	In Cluster 1 (fatigue, functional status, mood disturbances, and depression), the prevalence of moderate to severe symptoms one week after chemotherapy was: fatigue: 60.3%; impaired functional status: 47.6%; mood disturbances: 36.5%; depression: 18.5%. Adolescents diagnosed with solid tumors and those who had received prior chemotherapy were more likely to experience fatigue and depression (*p* < 0.01). In Cluster 2 (vomiting, nausea, sleep disturbance, and performance status), the prevalence of moderate to severe symptoms was: nausea: 42.4%; vomiting: 20.6%; altered performance status: 13.8%.
Lopes-Junior et al., 2018 [36] Brazil	Quasi-experimental 2B	To examine the feasibility of longitudinal testing of stress and fatigue in pediatric patients with osteosarcoma under chemotherapy who underwent clown intervention	6 children/adolescents with osteosarcoma (12.33 ± 3.32)	Symptoms: Cancer-related fatigue and psychological stress Instruments: CSS/ PedsQL MFS (self and proxy version)	Cortisol levels exhibited a decreasing trend over time in all six pediatric osteosarcoma patients. A similar trend was observed for tumor necrosis factor-α (TNF-α) levels across all six patients. In patients with metastatic osteosarcoma, metalloproteinase-9 levels showed a linear decrease between 1 and 9 h after the clown intervention.
Li et al., 2020 [37] China	Cross-sectional 3B	To investigate CSC in children with acute lymphoblastic leukemia receiving chemotherapy and their predictors	159 patients (97 boys and 62 girls) Mean age: 8.92 ± 3.33 years	Symptoms: Gastrointestinal, emotional, cognitive-related, self-image disorder symptoms, skin irritation cluster and somatic cluster Instrument: MSAS	Six CSC were identified, including the following: (i) gastrointestinal cluster; (ii) emotional cluster; (iii) cognitive cluster; (iv) self-image disorder cluster; (v) skin irritation cluster; and (vi) somatic cluster. Multiple linear regression analysis revealed that “chemotherapy phase, sex, and age were significantly associated with cluster severity, explaining 9.1–28.7% of the variance across clusters”.
Lopes-Junior et al., 2020 [38] Brazil	Quasi-experimental 2B	To evaluate the effect of clown theater art intervention on levels of stress and fatigue in pediatric cancer patients undergoing chemotherapy	16 children/adolescents with cancer	Symptoms: Cancer-related fatigue and psychological stress Instruments: CSS/ PedsQL MFS (self and proxy version)	Compared to baseline measurements, total psychological stress and cancer-related fatigue levels showed significant improvement following the clown intervention (*p* = 0.003 and *p* = 0.04, respectively). Additionally, salivary cortisol levels significantly decreased after the intervention at +1, +9, and +13 h (*p* < 0.05). In contrast, α-amylase levels remained unchanged.
Hooke et al., 2022 [39] United States	Longitudinal study 2A	To examine differences between CSC and quality of life (QoL) throughout the course of maintenance therapy for ALL	42 children and adolescents with cancer	Symptoms: Fatigue, sleep disturbance, and depression Instruments: CFS/AFS PROMIS/ GLTEQ	Two latent CSC classes (low and high) showed significant differences in symptoms and QoL at the start and end of maintenance therapy (*p* < 0.01), though both remained stable over time. Children with lower symptom burden and higher physical activity at baseline had better QoL outcomes post-treatment (*p* < 0.01).
Li et al., 2022 [40] China	Cross-sectional 3B	To examine CSC that children with ALL undergoing chemotherapy have which impact on their QoL	184 children, with the mean age of 10.38 (2.22) years	Symptoms: Gastrointestinal, emotional, neurological, skin mucosal, self-image disorder, and somatic cluster Instruments:MSAS/PedsQL	The severity of all 6 CSC was negatively correlated with QoL in children with ALL. Gastrointestinal, emotional, and somatic symptom clusters were the primary factors impacting QoL in children with ALL undergoing chemotherapy.
Wang et al., 2025 [41] China	Cross-sectional 3B	To explore the dynamic changes in symptom clusters among children with ALL during chemotherapy using electronic nursing records	75 children with ALL with an average age of 6.2 years	Symptoms: Cough, vomiting, fatigue, and fever Instruments: MSAS/PedsQL Generic Core Module v. 4.0)	The most common symptoms reported were bleeding, cough, and vomiting. Notably, upper gastrointestinal, respiratory, lower gastrointestinal, and skin-related symptom clusters persisted throughout both the consolidation and maintenance periods. Additionally, neurological and other discomfort-related symptom clusters were observed exclusively during the consolidation phase.
Li et al., 2024 [42] China	Longitudinal study 2A	To assess the stability of CSC in children with ALL during chemotherapy	134 children (8–16 years old), mean age 10.53 (±2.18) years	Symptoms: Gastrointestinal, emotional, neurological, skin mucosal, self-image disorder, and somatic cluster Instrument: MSAS	Six CSCs were identified. Emotional and somatic clusters were consistently present across all dimensions and time points. Gastrointestinal and self-image disorder clusters appeared at most time points, while neurological and skin–mucosa clusters were observed primarily at T2 and T3.

Abbreviations: CSC: cancer symptom clusters; PedsQL MFS: Pediatric Quality of Life Multidimensional Fatigue Scale; CFS: Child Fatigue Scale; AFS: Adolescent Fatigue Scale; PFS: Parental Fatigue Scale; CDI: Child Depression Inventory; BASC: Behavioral Assessment System for Children; LPPS: Lansky Play-Performance Scale; KPS: Karnofsky Performance Status Scale; MSAS: Memorial Symptom Assessment Scale; CCS: Child Stress Scale; PROMIS: Patient-Reported Outcomes Measurement Information System; GLTEQ: Godin Leisure-Time Exercise Questionnaire; QoL: Quality of Life. ALL: acute lymphoblastic leukemia.

**Table 3 nursrep-15-00163-t003:** ROBINS-I consensus assessment of quasi-experimental studies.

	ROBINS-I * Domains							Overall Judgment ROBINS-I *
Study	Confounding Bias	Selection Bias	Intervention Classification Bias	Bias Due to Intervention Deviations	Incomplete Data Bias	Outcome Measurement Bias	Selective Outcome Reporting Bias	
Hockenberry et al., 2010 [31]	Low	Low	Low	Low	Low	Low	Low	Low
Hockenberry et al., 2011 [35]	Low	Low	Low	Low	Low	Low	Low	Low
Lopes-Junior et al., 2018 [36]	Low	Low	Low	Low	Low	Low	Low	Low
Lopes-Junior et al., 2020 [38]	Low	Moderate	Low	Low	Low	Low	Low	Moderate

Abbreviations: * ROBINS-I, Risk of Bias in Non-randomized Studies.

**Table 4 nursrep-15-00163-t004:** JBI checklist for analytical cross-sectional studies.

JBI Critical Appraisal Checklist	Were the Inclusion Criteria for the Sample Clearly Defined?	Were the Study Participants and the Study Location Described in Detail?	Was the Exposure Measured Reliably and Validly?	Were Objective and Standardized Criteria Used to Assess Health Status?	Were Confounding Factors Identified and Addressed?	Have Strategies Been Established to Manage Confounding Factors?	Were the Outcomes Measured Reliably and Validly?	Was the Appropriate Statistical Analysis Applied?
Atay, 2011 [30]	Yes	Yes	Yes	Yes	No	No	Yes	Yes
Atay, 2012 [31]	Yes	Yes	Yes	Yes	Yes	Yes	Yes	Yes
Baggott et al., 2012 [34]	Yes	Yes	Yes	Yes	Yes	Yes	Yes	Yes
Li et al., 2020 [37]	Yes	Yes	Yes	Yes	Yes	Yes	Yes	Yes
Hooke et al., 2022 [39]	Yes	Yes	Yes	Yes	Yes	Yes	Yes	Yes
Rongrong Li et al., 2022 [40]	Yes	Yes	Yes	Yes	Yes	Yes	Yes	Yes
Wang et al., 2025 [41]	Yes	Yes	Yes	Yes	Yes	Yes	Yes	Yes
Rongrong Li et al., 2024 [42]	Yes	Yes	Yes	Yes	Yes	Yes	Yes	Yes

## Data Availability

Data are available upon reasonable request.

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
