# Peer review of "Cancer Symptom Clusters in Children and Adolescents with Cancer Undergoing Chemotherapy: A Systematic Review"

_nursrep, 2025, doi:10.3390/nursrep15050163_

Round 1
Reviewer 1 Report
Comments and Suggestions for Authors
Dear Editor,
I appreciate the opportunity to review the article titled "Cancer Symptom Cluster in Children and Adolescents with Cancer Under Chemotherapy: A Systematic Review." The aim of this review was to synthesize and analyze the prevalence, composition, longitudinal stability, and predictors of cancer symptom clusters in children and adolescents undergoing chemotherapy.
The topic is indeed intriguing. However, the fact that the research was conducted in December 2022, as noted in the abstract, suggests that an update is necessary. I recommend that the authors update their research to incorporate the most recent studies and revise their manuscript accordingly.
Introduction: It is advisable to update the manuscript with the most recent information: Cancer in children and adolescents accounts for 1 to 4% of all malignant tumors in most populations. In low- and middle-income countries, where the pediatric population represents approximately 50%, this proportion is estimated to range between 3 and 10% of all neoplasms.
Language Consistency: There are phrases in Portuguese that need correction or translation. For example: "One of the challenges for Pediatric Oncology Nursing is indeed the exploration of these oncological symptoms to clearly demonstrate the significance of these clusters in terms of interaction levels, association patterns, and synergy."
Literature Review: The assertion that there are no reviews on the topic seems inaccurate as several studies have addressed similar themes:
- Kamkhoad, D., Santacroce, S. J., Phonyiam, R., & Mian Wang. (2023). "Symptom Clusters That Included Gastrointestinal Symptoms Among Children Receiving Cancer Treatments: A Scoping Review." Oncology Nursing Forum, 50(3), 381–395. https://doi.org/10.1188/23.ONF.381-395
- Tomlinson D, Tigelaar L, Hyslop S, et al. (2017). "Self-report of symptoms in children with cancer younger than 8 years of age: a systematic review." Supportive Care in Cancer, 25(8): 2663-2670. doi:10.1007/s00520-017-3740-6
- Erickson, Jeanne M., et al. (2013). "Symptoms and Symptom Clusters in Adolescents Receiving Cancer Treatment: A Review of the Literature." International Journal of Nursing Studies, 50(6): 847–69. doi:10.1016/j.ijnurstu.2012.10.011
Methodology: Do observational studies adequately address the systematic review and the PICO question? This aspect needs reassessment, particularly in terms of methodological rigor and relevance.
Results: The assertion that the two most prevalent cancer symptom clusters among children and adolescents undergoing chemotherapy are the neuropsychological and gastrointestinal clusters is questionable based on the data presented. I recommend re-extracting the data, including articles from 2023 and 2024, to ensure the findings are accurate and current.
Overall, revising these areas will significantly enhance the credibility and utility of the review, providing clearer insights into the symptom clusters affecting children and adolescents undergoing chemotherapy.
Best regards,
Author Response
Comments and Suggestions for Authors
Dear
I appreciate the opportunity to review the article titled "Cancer Symptom Cluster in Children and Adolescents with Cancer Under Chemotherapy: A Systematic Review." The aim of this review was to synthesize and analyze the prevalence, composition, longitudinal stability, and predictors of cancer symptom clusters in children and adolescents undergoing chemotherapy.
The topic is indeed intriguing. However, the fact that the research was conducted in December 2022, as noted in the abstract, suggests that an update is necessary. I recommend that the authors update their research to incorporate the most recent studies and revise their manuscript accordingly.
Response: The systematic review has been updated. A new search was conducted in February 2025, resulting in the inclusion of four additional studies published between 2021 and 2024. The manuscript has been revised accordingly to reflect these updates.
Introduction: It is advisable to update the manuscript with the most recent information: Cancer in children and adolescents accounts for 1 to 4% of all malignant tumors in most populations. In low- and middle-income countries, where the pediatric population represents approximately 50%, this proportion is estimated to range between 3 and 10% of all neoplasms.
Response: We have updated the introduction with the latest epidemiological data on childhood and adolescent cancer, incorporating statistics on its prevalence in low- and middle-income countries.
Language Consistency: There are phrases in Portuguese that need correction or translation. For example: "One of the challenges for Pediatric Oncology Nursing is indeed the exploration of these oncological symptoms to clearly demonstrate the significance of these clusters in terms of interaction levels, association patterns, and synergy."
Response: All instances of non-English text have been corrected. The entire manuscript has undergone a full linguistic review by a native English speaker.
Literature Review: The assertion that there are no reviews on the topic seems inaccurate as several studies have addressed similar themes:
- Kamkhoad, D., Santacroce, S. J., Phonyiam, R., & Mian Wang. (2023). "Symptom Clusters That Included Gastrointestinal Symptoms Among Children Receiving Cancer Treatments: A Scoping Review." Oncology Nursing Forum, 50(3), 381–395. https://doi.org/10.1188/23.ONF.381-395
- Tomlinson D, Tigelaar L, Hyslop S, et al. (2017). "Self-report of symptoms in children with cancer younger than 8 years of age: a systematic review." Supportive Care in Cancer, 25(8): 2663-2670. doi:10.1007/s00520-017-3740-6
- Erickson, Jeanne M., et al. (2013). "Symptoms and Symptom Clusters in Adolescents Receiving Cancer Treatment: A Review of the Literature." International Journal of Nursing Studies, 50(6): 847–69. doi:10.1016/j.ijnurstu.2012.10.011
Response: We acknowledge this point and have revised the literature review to clarify that while previous reviews exist, none have comprehensively synthesized evidence on the prevalence, composition, longitudinal stability, and predictors of symptom clusters in pediatric oncology. Our study fills this gap by providing a broader and more systematic analysis.
We have rewritten the rationale for clarity. “Despite growing interest in understanding symptom clusters in oncology, existing systematic reviews have primarily focused on the adult population (16-17), or have provided a partial perspective on pediatric oncology. While previous systematic reviews have explored aspects of symptom clusters in specific pediatric cancer contexts, they have not comprehensively synthesized evidence on the prevalence, composition, longitudinal stability, and predictors of symptom clusters in children and adolescents undergoing chemotherapy. For instance, Kamkhoad et al. (2023) conducted a scoping review focusing on gastrointestinal symptom clusters in pediatric oncology, providing valuable insights but lacking a systematic synthesis of broader symptom clusters and their interrelationships. Similarly, Tomlinson et al. (2017) reviewed self-reported symptoms in children under 8 years of age, but their scope was limited to younger pediatric patients and did not analyze symptom clustering patterns in a broader age range. Additionally, Erickson et al. (2013) reviewed symptom clusters in adolescents, but their study was not a systematic review, nor did it examine longitudinal stability or predictive factors that influence symptom cluster trajectories.
Given these limitations, the present systematic review fills a critical gap by synthesizing comprehensive evidence on cancer symptom clusters in pediatric and adolescent populations. Understanding how these clusters evolve over time and identifying predictive factors for co-occurring symptoms could provide a scientific foundation for developing personalized therapeutic interventions. Furthermore, this approach may facilitate the early identification of symptom patterns predictive of significant symptom clusters, ultimately contributing to improved symptom management and better quality of life during and after cancer treatment.
Methodology: Do observational studies adequately address the systematic review and the PICO question? This aspect needs reassessment, particularly in terms of methodological rigor and relevance.
Response: We have reassessed our methodology and adjusted the PICO framework to PECO, which is more suitable for this type of systematic review. The revised methodology now ensures greater alignment with our research objectives.
Results: The assertion that the two most prevalent cancer symptom clusters among children and adolescents undergoing chemotherapy are the neuropsychological and gastrointestinal clusters is questionable based on the data presented. I recommend re-extracting the data, including articles from 2023 and 2024, to ensure the findings are accurate and current.
Response: We have re-extracted and analyzed the data, incorporating newly identified studies. Based on the updated analysis, we have modified the description of the most prevalent symptom clusters. The revised results now highlight the gastrointestinal, emotional, fatigue, somatic, and self-image clusters as the most prevalent. Additionally, we have refined our conclusion to reflect these findings accurately.
Updated Conclusion:
“Findings from this systematic review indicate that cancer symptom clusters in children and adolescents follow well-defined patterns, with the gastrointestinal, emotional, fatigue, somatic, and self-image clusters being the most prevalent. Identifying these clusters is essential for developing targeted interventions to reduce symptom burden and improve quality of life in pediatric oncology patients. The longitudinal stability of cancer symptom clusters varies depending on disease stage and treatment phase, with inflammatory mediators, particularly pro-inflammatory and anti-inflammatory cytokines, serving as key predictors of symptom cluster configurations. Early identification of these cancer symptom clusters is critical for guiding interdisciplinary teams in delivering personalized, patient-centered care for children and adolescents with cancer and their families.”
Overall, revising these areas will significantly enhance the credibility and utility of the review, providing clearer insights into the symptom clusters affecting children and adolescents undergoing chemotherapy.
Response: Thank you very much for your constructive feedback. We have carefully addressed all the suggested revisions and believe these improvements have strengthened our study. Best regards,
Reviewer 2 Report
Comments and Suggestions for Authors
Authors have performed a systematic review to evaluate the presence of cancer symptom cluster in children and adolescents with cancer under chemotherapy. This study is of interest to readers as oncological population may face many different symptoms during and after medical treatment, and it is important to define them adequately so clinicians can adapt the interventions individually.
However, authors have to revise their manuscript as there are some sentences that are not written in English, as seen in page 2, lines 45-48.
Also, I would recommend authors to be more specific in tables and figures, as there is a huge amount of text and is difficult to read. Moreover, rather than figures, I would name these (figures 1,3,4,5) as tables instead of figures.
Comments on the Quality of English Language
Some sentences have to be modified, as they are not written in English.
Author Response
Comments and Suggestions for Authors
Authors have performed a systematic review to evaluate the presence of cancer symptom cluster in children and adolescents with cancer under chemotherapy. This study is of interest to readers as oncological population may face many different symptoms during and after medical treatment, and it is important to define them adequately so clinicians can adapt the interventions individually.
Response: Thank you very much for your positive feedback regarding our study. We appreciate your recognition of its significance.
However, authors have to revise their manuscript as there are some sentences that are not written in English, as seen in page 2, lines 45-48.
Response: We have corrected these sentences and ensured that the entire manuscript has been reviewed for language accuracy. Thank you for pointing this out.
Also, I would recommend authors to be more specific in tables and figures, as there is a huge amount of text and is difficult to read. Moreover, rather than figures, I would name these (figures 1,3,4,5) as tables instead of figures.
Response: We have revised the tables and figures to improve readability and clarity. Additionally, we have renamed the relevant figures as tables, as suggested. Thank you for your valuable input.
Comments on the Quality of English Language
Some sentences have to be modified, as they are not written in English.
Response: The entire manuscript has undergone thorough revision by a native English speaker to ensure linguistic accuracy and adherence to academic standards.
Round 2
Reviewer 1 Report
Comments and Suggestions for Authors
Dear Editor and Reviewer,
I hope this message finds you both well!
I want to congratulate the authors for the latest revisions submitted for the article Cancer symptom cluster in children and adolescents with cancer under chemotherapy: a systematic review." I am pleased with the changes made and believe they significantly enhance the quality of the work.
However, I would like to request one additional minor modification: would it be possible to try to reduce the text in Table 2 a bit? I believe a more concise presentation could facilitate the understanding of the data.
Thank you for your attention, and I look forward to your thoughts.
Sincerely,
CSF
Author Response
Dear Reviewer,
We have accepted your suggestion and significantly reduced Table 2 while preserving the key information. The revised version has greatly improved readability—reducing the original four pages to two. All changes have been highlighted in yellow. We are grateful for this recommendation, as well as for all your careful and constructive contributions to enhance our manuscript.
Sincerely,
Dr. Lopes-Júnior